# The Effects of Alexithymia and Confidence in Patient Safety Management on the Clinical Competence of Nursing Students: A Cross-Sectional Survey

**DOI:** 10.3390/healthcare11182504

**Published:** 2023-09-09

**Authors:** KyoungSook Lee, YoungSook Kim

**Affiliations:** 1Department of Nursing, University of Ulsan, Ulsan 44610, Republic of Korea; gslee1@ulsan.ac.kr; 2Department of Nursing, University of Uiduk, Gyeongju 38004, Republic of Korea

**Keywords:** clinical competence, patient safety, students, nursing

## Abstract

Nursing students frequently believe that they are clinically incompetent. The objective of this study was to identify the effects of alexithymia and patient safety management on the clinical competence of nursing students. A cross-sectional design was used to investigate these relationships among 167 nursing students from two universities in South Korea. A self-reported structured questionnaire was used for alexithymia, patient safety management confidence, and clinical competence. The factors influencing clinical competence were safety education (*β* = −0.16, *p* = 0.022), alexithymia (*β* = −0.14, *p* = 0.049), and confidence in patient safety management (*β* = 0.52, *p* < 0.001). The explanatory power of these three factors was 37.1%, and confidence in patient safety management was found to have the greatest influence on clinical competence. Based on these results, the pursuit of patient safety and the delivery of high-quality care depend not only on the acquisition of clinical skills but also on the emotional competencies and patient safety management confidence of the nursing students.

## 1. Introduction

The goal of nursing education is to prepare nurses to deal with various clinical situations by increasing nursing expertise and clinical competence [1]. The clinical competence of nursing students is an important factor in determining their ability to independently perform nursing tasks professionally after graduation [2]. Nursing competence refers to the abilities required to accomplish the role of nurses [3]. Because the clinical competence of nursing students is built by handling various clinical cases, clinical practice potentially plays an important role in improving clinical performance [4]. It is critical to improve the clinical competence of nursing students because higher clinical competence equals higher professionalism in nursing and adaptability in the clinical field [5]. In particular, during the challenging situation of the COVID-19 pandemic, improvement in clinical competence of nurses and nursing students is more important than ever [5]. Nursing students will have a chance to learn communication skills and methods to communicate with medical staff and subjects in various nursing situations in clinical practice [6]. Nursing students can empathize with patients via efficient communication and relationship formation in clinical practice [7]. However, alexithymia hinders the development of empathy. Clear communication without expressing emotions is critical in complex and busy clinical fields [8] as expressing emotions disrupts clear communication, resulting in patient safety accidents such as crisis discovery, medication errors, delayed diagnosis, and death [9]. Since nursing students must play the role of medical personnel as prospective nurses, it is essential for them to have the expertise and skills to improve the awareness of patient safety and provide safe nursing [10]. Clinical competence, a cornerstone of effective nursing practice, encompasses not only the application of medical knowledge but also the ability to communicate effectively, exhibit emotional intelligence, and ensure patient safety. In recent years, researchers and educators have shown an increased interest in understanding the factors that influence the clinical competence of nursing students.

One emotional factor that has garnered increased attention in recent years is alexithymia, a personality trait characterized by difficulties in recognizing, understanding, and expressing emotions [11]. Effective patient care often involves not only technical skills but also empathy, communication, and understanding of patient emotions. It is posited that nursing students with higher levels of alexithymia may encounter challenges in connecting with patients on an emotional level, potentially affecting their clinical competence [12,13,14].

Confidence in patient safety management plays a pivotal role in ensuring the well-being of patients. Confidence in this context encompasses the belief of nursing students in their capacity to maintain patient safety, prevent medical errors, and effectively manage adverse events. While confidence is often viewed as a significant component of clinical competence, the extent to which alexithymia may interact with confidence in patient safety management and ultimately affect the clinical competence of nursing students remains a subject worthy of exploration [15,16,17].

The general curriculum of nursing involves gradual sequences. First-grade students study general elective subjects, and second-grade students study basic major subjects and fundamental nursing practices in school. Then, third-grade students study advanced major subjects and clinical practices. Clinical practice-based education helps them to acquire more skills, where they are exposed to various clinical practices, adapt to the role of nursing, and acquire nursing skills [17].

Previous research has found that nursing students in Korea who are about to graduate have lesser knowledge about patient safety than nursing students in other countries and have moderate confidence in their abilities [18].Furthermore, one study discovered that although 81.6% of students received patient safety education, their self-awareness was only moderate [19]. Research is being conducted on clinical competence [17], clinical practice satisfaction, clinical stress, patient safety education [19], and communication skills [20,21], as well as various studies related to subjects affecting patient safety management confidence [18,22] on nursing students who have had clinical experience. However, current observation-oriented clinical practice has difficulty promoting clinical competence while also limiting clinical competence development through clinical practice [16].

The key to nursing education is to encourage complete clinical competence among nursing students. Furthermore, the modifiable variables that affect nursing students’ clinical competence must be identified given the need to ensure clinical competence through nursing education [18]. In particular, nursing students may experience clinical maladaptation as a result of alexithymia [13]. Furthermore, they acknowledged that they have low confidence in safety management [22], which is an important internal factor that can negatively affect clinical performance. The situation in 2020, when clinical practice in the field was limited, perhaps caused nursing students’ confidence in their clinical competence, alexithymia, and safety management to suffer.

This study seeks to investigate the intricate relationships between alexithymia, confidence in patient safety management, and the clinical competence of nursing students. By examining these variables, we aim to provide valuable insights into how emotional factors, particularly alexithymia, may intersect with the development of nursing students’ clinical competence and their ability to ensure patient safety in healthcare settings.

## 2. Methods

### 2.1. Study Design

The study adopted a cross-sectional study to determine how alexithymia and patient safety management affect the clinical competence of nursing students (Figure 1).

### 2.2. Study Participants

This study used a convenience sampling method to recruit nursing students from two universities in the U metropolitan city and the G city. The number of samples was determined using the G*Power 3.1 program and confirmed using the following parameters needed for the regression analysis: 0.05 significance level, 0.90 explanatory power, 0.15 medium effect size, and10predictor variables. Following that, we determined that a total of 142 samples were required. Despite a dropout rate of approximately 20%, 170 students were enrolled in this study. Thereafter, 170 copies were collected and 167 people were targeted, excluding 3 who provided insufficient responses. The response rate was 98.2%. The following are the detailed acceptance criteria for universities and participants: a university must (1) be located in the U or G city and (2) have a nursing department head who understands the purpose of the study and allows the questionnaire to be completed. Participants could come from either of the two selected universities. They spoke Korean fluently, were able and willing to complete the questionnaire, and provided informed consent.

For ethical considerations regarding the patients, researchers and research investigators personally visited each university, explained the purpose and procedure of the study to each professor, obtained permission, and collected data only from students who wished to participate in the study with the cooperation of the professor. Before the study, participants were asked to read enough of the consent form to participate in the survey and voluntarily participate in the survey, and they were informed in writing that they could cancel their participation at any time if they did not want to proceed. Students who completed the consent form and submitted the questionnaire received a small token.

### 2.3. Study Population

There were two universities in the U metropolitan city and the G city. The questionnaire was sent to 50% of the third- and fourth-grade nursing students of these universities.

### 2.4. Measures

#### 2.4.1. Variables

Structured self-report questionnaires comprising 86 questions, including 8 on general characteristics, 23 on alexithymia, 10 on confidence in patient safety management, and 45 on clinical competence, were used in this study.

#### 2.4.2. Alexithymia

The Bag by, Parker, and Taylor alexithymia scale (20-Item Toronto Alexithymia Scale (TAS-20) [23] and the Shin and Won emotional expression inability scale (TAS-20K) [20] were used. A total of 23 questions were measured on a Likert 5-point scale, and the total score was calculated using three sub-factors: the ability to check emotions and distinguish body sensations about emotions (or difficulty in identifying feelings), outward-oriented thinking (or externally oriented thinking), and the ability to explain emotions to others (or difficulty in describing feelings). A total of 23 questions were scored on a Likert 5-point scale of 0–4 points (0: strongly disagree, 1: disagree, 2: neutral, 3: agree, 4: strongly agree), and the total score was calculated using three sub-scale scores: difficulty in identifying feelings, difficulty in describing feelings, and externally oriented thinking. Negative questions were scored in reverse. Cronbach’s α was 0.81 [19] during tool development and 0.76 during the transition [24]. Regarding reliability, the Korean version had a Cronbach’s α of 0.82, whereas this study had a Cronbach’s α of 0.86.

#### 2.4.3. Confidence in Patient Safety Management

To confirm confidence in patient safety management, Madigosky, Headrick, Nelson, Cox, and Anderson [25] developed a tool to confirm confidence in patient safety management for medical students, and patient safety assessment tools [18], translated by Park et al. [18], were used to measure patient safety scores by referring to international patient safety management goals. This tool employs a Likert 5-point scale (1: strongly disagree, 2: disagree, 3: neutral, 4: agree, and 5: strongly agree), with higher scores across the 10 questions indicating a greater confidence in the performance of patient safety management. Negative questions were scored in a reverse order. Cronbach’s α was 0.85 in the studies by Madigosky, Headrick, Nelson, Cox, and Anderson [25], 0.85 in the study by Park [18], and 0.83 in the current study.

#### 2.4.4. Clinical Competence

To assess clinical performance in this study, the tools developed by Lee et al. [26] and modified and supplemented by Choi [27] were used. The tool consists of 45 questions scored on a 5-point Likert scale (1: strongly disagree, 2: disagree, 3: neutral, 4: agree, and 5: strongly agree), with higher scores indicating better clinical performance. Negative questions were scored in a reverse order. The total score was calculated by combining the scores obtained in five subscales: nursing process, nursing skill, education and education/cooperation, interpersonal/communication, and professional development. The developed tool had a Cronbach’s α of 0.96 [26], while the modified tool by Choi [27] had a Cronbach’s α of 0.90. In this study, Cronbach’s α was 0.89.

### 2.5. Ethical Considerations

This study was carried out after the institutional review board of A university (IRB No.: IRB/GU-2037) for research subject rights and ethical considerations.

### 2.6. Data Collection

This study was carried out after the institutional review board of A university (IRB No.: IRB/GU-2037) approved the research design and methods. The participants included two universities from the U metropolitan city and the G city, which received official cooperation and approval. Data were collected via a questionnaire survey from 1 October 2020 to 15 October 2020. Before completing the questionnaire, the study participants were given an explanation of the study’s objectives, the data collection method, and the strict use of the data collected for only the purposes stated. The researcher went to the chosen university and handed out the questionnaire to third-and fourth-grade nursing students. After completing the questionnaires, the researcher went to collect them herself. Following the class, the questionnaire was distributed to the appropriate subjects in the classroom, and students who completed it were asked to place it in a sealed bag. The survey took between 10 and 20 min to complete. The researcher organized and analyzed the data without attempting to track down or identify the participants. The consent form and survey data will be kept for 3 years before being destroyed. Furthermore, the coded data will be retained for at least 5 years before being deleted. A total of 170 questionnaire copies were distributed and collected from those who agreed to participate in this study. However, three respondents who responded incompletely were disqualified. Finally, 167 responses were scrutinized.

### 2.7. Data Analysis

Data analysis was carried out using SPSS for Windows (Version 25.0). The general characteristics of the participants were defined using numbers and percentages. Alexithymia, confidence in patient safety management, and clinical competence were expressed as the mean (standard deviation). The study participants’ clinical competence was assessed based on their general characteristics using the independent *t*-test. Before the *t*-test analysis, normality was verified to confirm the inclusion of normality. The association between alexithymia, confidence in patient safety management, and clinical competence was examined using Pearson’s correlation coefficient. The effects of alexithymia and confidence in patient safety management on clinical competence were investigated using hierarchical regression analysis. Before the hierarchical regression analysis, normality was verified to confirm the inclusion of normality. A *p*-value of <0.05 denoted statistical significance.

## 3. Results

### 3.1. Clinical Competence Based on General Characteristics

Table 1 shows that the participant’s clinical competence varied significantly by grade (t = −2.09, *p* < 0.001), major satisfaction (t = 2.25, *p* = 0.026), satisfaction with clinical practice (t = 2.35, *p* = 0.020), safety education (t = 2.23, *p* = 0.027), and subjective health status (t = 2.64, *p* = 0.019). Among the general characteristics, sex, scholastic performance, and religion were not significantly correlated with clinical competence. Table 1 shows the specific characteristics.

According to Table 1, this study included 167 nursing students, of whom 123 (73.7%) were female. Regarding their grade, 88 participants (52.7%) were in fourth. Regarding academic performance, 99 participants (59.3%) had an average academic score below 3.0. Regarding major satisfaction, 89 participants (53.3%) responded with “satisfaction”. Regarding satisfaction with clinical practice, 86 participants (51.5%) responded with “satisfaction”. Regarding safety education, 146 participants (87.5%) said “Have”. Regarding religion, 118 participants (70.7%) replied “Have”. In terms of subjective health status, 85 participants (50.9%) responded they were “moderate or poor” (Table 1).

### 3.2. Study Subjects’ Alexithymia, Confidence in Patient Safety Management, and Clinical Competence

The mean scores of participants’ alexithymia, confidence in patient safety management, and clinical competence are mentioned in Table 2. The participants’ average alexithymia score mean ± standard deviation (SD) was 2.47 ± 0.49. The participants’ average confidence patient safety management score mean ± SD was 4.23 ± 0.40. The participants’ average clinical competence score mean ± SD was 4.01 ± 0.51.

### 3.3. Correlations between Variables

Table 3 shows the relationships between alexithymia, confidence in patient safety management, and clinical competence. Alexithymia was negatively correlated with confidence in patient safety management (r = −0.35, *p* < 0.001) and clinical competence (r = −0.35, *p* < 0.001), whereas confidence in patient safety management was positively correlated with clinical competence (r = 0.59, *p* < 0.001).

### 3.4. Factors Influencing Clinical Competence

To identify the factors influencing participants’ clinical competence, hierarchical regression analysis was used, the results of which are shown in Table 4.

A multiple regression analysis was used to examine general characteristics such as year, major satisfaction, satisfaction with clinical practice, safety education, subjective health status, alexithymia, and confidence in patient safety management. The assumption of a normal sample distribution was validated using absolute values of skewness and kurtosis. As a result of checking the Z-value of the univariate normality in this study, none of the measured variables exceeded the absolute values of skewness 3, kurtosis 7, and Z-value. As a result, it was determined that no specific issue existed in the sample normality review.

Among the predictor variables, the variable measured on a nominal scale was designated as the dummy variable. To test the residual independence, the Durbin–Watson test statistic was calculated, which was 1.99, and no auto-correlation was discovered for error terms. Therefore, residual independence was achieved. The tolerance of each variable was 0.43–0.89, indicating that no value was less than 0.10. The variance inflation factor of each variable was 1.12–2.31, which was less than 10; thus, the variables were not multicollinearity.

The regression model of Model 1 was significant (F = 3.51, *p* < 0.001). Among the general characteristics of the study subjects, grade (*β* = 0.24, *p* = 0.007), major satisfaction (*β* = −0.19, *p* = 0.046), safety education (*β* = −0.16, *p* = 0.049), and subjective health status (*β* = −0.19, *p* = 0.027) had a significant impact on clinical competence. The explanatory power of the four factors was 12.2%.

The regression model of Model 2 was significant (F = 13.14, *p* < 0.001). The factors influencing clinical competence were safety education (*β* = −0.13, *p* = 0.031) and alexithymia (*β* = −0.34, *p* < 0.001), which had a significant impact on clinical competence. The explanatory power of the two factors was 14.3%.

The regression model of Model 3was significant (F = 13.14, *p* < 0.001). The factors influencing clinical competence were safety education (*β* = −0.16, *p* = 0.022), alexithymia (*β* = −0.14, *p* = 0.049), and confidence in patient safety management (*β* = 0.52, *p* < 0.001). The explanatory power of these three factors was 37.1%, and confidence in patient safety management was found to have the greatest influence on the clinical competence.

## 4. Discussion

This study was conducted to identify the factors that affect the clinical competence of nursing students. The first regression model identified the general characteristics affecting the clinical competence of nursing students. Fourth graders had higher clinical competence than third graders, and subjects who reported having major satisfaction were more satisfied than those who reported having moderate satisfaction or dissatisfaction. Subjects who were satisfied with clinical practice scored higher than those who were moderately satisfied or dissatisfied with clinical practice. Subjects with good health status outperformed those with moderate or poor health status in terms of clinical performance.

Our results were similar to those obtained in previous studies [23,28] that reported that fourth graders exhibit better clinical performance than third graders. We obtained another result that was similar to previous studies [29,30], which indicated that major satisfaction and clinical practice satisfaction affected clinical competence. The average clinical competence of participants in this study was 4.01 out of 5, which was comparable to that reported in a study (higher than 3.54) [28] that used the same tool as this study to assess third and fourth graders from the same nursing college [29]. Major satisfaction is positively correlated with clinical competence [30]. The improvement in clinical competence observed in this study with an increase in the grade might be attributed to better education and clinical practice. Efforts should be made to increase major satisfaction, as the group with major satisfaction had higher clinical competence than the ordinary but dissatisfied groups. In addition, in this study, participants who stated that their subjective health status was good had higher clinical competence than those who said that it was normal or bad; therefore, systems and education will be needed to improve the subjective health status of such individuals.

In the second regression model, the factors influencing clinical competence were safety, alexithymia, and education order. In the third regression model, factors influencing clinical competence were confidence in patient safety, alexithymia, and safety education order. Notably, our findings revealed that the most important factor influencing clinical competence was confidence in patient safety management. The lack of studies on whether confidence in patient safety management affects clinical competence among nursing college students made comparisons difficult. However, a study on nursing students [19] found that patient safety knowledge and attitudes are significantly correlated with performance confidence, which supports our findings. Patient safety incidents cause psychological harm and pain to patients, prolong hospitalization, and aggravate pre-existing diseases, resulting in financial losses and jeopardizing the patient’s condition [31]. Therefore, preventing patient safety accidents by improving patient safety management capabilities is essential not only for medical institution workers but also for nursing students participating in patient nursing in clinical practice.

The second-most important factor influencing clinical competence in the current study was safety education. It was difficult to make a direct comparison of these results due to the small number of studies that included safety education as one of the factors influencing the clinical competency of nursing students. As safety education influences confidence in patient safety management, a study of nursing students found that patient safety knowledge and attitudes were significantly correlated with performance confidence [22]. In our study, 87.5% of nursing students reported receiving patient safety management education, which is higher than the proportion (79.4%) reported in another study [28]. Issues in patient safety involve complex factors, such as hospital environment and patient condition; thus, convergence education is required to improve the capabilities of nursing students to manage patient safety.

In the current study, the third-most important factor influencing clinical competence was alexithymia. Few studies looked at the direct relationship or influence of alexithymia on clinical competence, making direct comparisons difficult. However, studies have shown that the most influential factor in the clinical performance ability of nursing college students was communication ability, which could apply to alexithymia [5,18]. Our findings revealed that the average alexithymia score of nursing students was 2.47 out of 4 points, which was similar to the 2.66 points reported in another study on nursing students using the same tool. The subjects of this study had moderate levels of alexithymia. Nursing students with high alexithymia scores are unable to correctly recognize other people’s emotions, resulting in various maladjustments such as emotion recognition and difficulty expressing themselves [32], as well as difficulty forming interpersonal relationships due to a lack of understanding and empathy for others [33]. Due to this maladjustment, alexithymia is thought to have a negative impact on the clinical competence of nursing students. Given that an increase in the tendency to be unable to express emotions reduces the ability to solve social problems [32], various approaches should be taken to promote healthy emotional expression among nursing students.

The subjects included herein were more satisfied with clinical practice than those included in previous studies [12,34]. Thus, it appears that satisfaction with clinical practice in this study did not impact clinical competence.

According to the above research findings, there is a close relationship between satisfaction with clinical practice and clinical competence of nursing students [31], and thus various educational strategies are required to increase clinical practice satisfaction and clinical competence.

The identification of a relationship between alexithymia and clinical competence in the absence of research on alexithymia of nursing college students is of great importance. Confidence in patient safety management was found to have the greatest influence on clinical competence. Therefore, it has been proven that safety management education cannot be overemphasized. There is a need to strengthen education on safety for nursing students. This study is significant for the following reasons: First, because this study was designed for third- and fourth-grade students, the degree of variability in nursing students can be predicted. Thus, by developing and implementing clinical competence programs, nursing students’ clinical competence can be improved. Such programs should be offered to nursing students with low confidence in patient safety management or a high alexithymia score. Second, because we discovered that safety management education influences clinical competence, it is critical to strengthen safety management education for nursing students before clinical practice.

Our study revealed a significant association between alexithymia and lower levels of clinical competence among nursing students. Students with higher levels of alexithymia tended to exhibit difficulties in emotional expression and recognition, which, in turn, affected their ability to navigate emotionally charged clinical situations effectively.

## 5. Conclusions and Limitations

This study has some limitations. First, our findings cannot be generalized because the study only included nursing students from specific regions. Second, in this study, alexithymia, confidence in patient safety management, and clinical competence were measured using a self-reported questionnaire only. In the future, it is necessary to collect data through observation or performance evaluation. We also propose an experimental study on whether a program that increases confidence in patient safety management and decreases alexithymia in nursing students affects clinical competence.

The study participants had a relatively high degree of alexithymia, confidence in patient safety management, and clinical competence. In addition, major satisfaction and practical satisfaction were high, and 87.5% had received safety education. Clinical competence according to general characteristics differed according to grade, major satisfaction, practice satisfaction, safety education, and subjective health status perception.

The current study found that students with high levels of confidence in patient safety management and low levels of alexithymia and students who received safety management education had high clinical competence. Confidence in patient safety management had the greatest influence on their clinical competence.

This study aims to provide valuable insights into the effects of alexithymia and confidence in patient safety management on the clinical competence of nursing students. The findings will contribute to nursing education and practice by guiding interventions to enhance the preparedness of future nurses to deliver safe and competent patient care.

## Figures and Tables

**Figure 1 healthcare-11-02504-f001:**
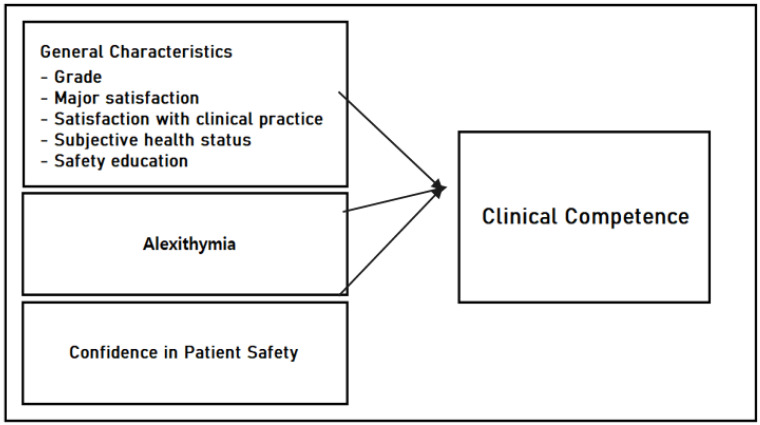
Conceptual framework.

**Table 1 healthcare-11-02504-t001:** Clinical competence based on general characteristics (*n* = 167).

Characteristics	Category	*n*(%)	Clinical Competence			
			Mean	SD	t	*p*
Sex	Female	123(73.7)	4.02	0.4	−0.71	0.481
	Male	44(26.3)	3.94	0.54		
Grade	3rd	79(47.3)	3.91	0.46	−2.09 *	0.039
	4th	88(52.7)	4.09	0.48		
Scholastic performance	<3.0	99(59.3)	4.06	0.43	1.85	0.066
	≥3.0	68(40.7)	3.89	0.46		
Major satisfaction	Satisfaction	89(53.3)	4.08	0.49	2.25 *	0.026
	Moderate or	78(46.7)	3.89	0.41		
	Dissatisfaction					
Satisfaction with clinical practice	Satisfaction	86(51.5)	4.02	0.49	2.35 *	0.02
	Moderate or	81(48.5)	3.88	0.41		
	Dissatisfaction					
Safety education	Have	146(87.5)	4.03	0.42	2.23 *	0.027
	Have not	21(12.5)	3.69	0.41		
Religion	Have	49(29.3)	4	0.44	0.054	0.957
	Have not	118(70.7)	4.01	0.51		
Subjective health status	Good	82(49.1)	4.09	0.46	2.64 *	0.019
	Moderate or	85(50.9)	3.9	0.46		
	Poor					

* *p* < 0.05.

**Table 2 healthcare-11-02504-t002:** Levels of alexithymia, confidence in patient safety management, and clinical competence (*n* = 167).

Variable	Mean	SD	Range	Min.	Max.
Alexithymia	2.47	0.49	0–4	1.38	3.55
Difficulty in identifying feelings	2.22	0.45	0–4	1.51	3.12
Difficulty in describing feelings	2.74	0.49	0–4	1.52	3.74
Externally oriented thinking	2.46	0.52	0–4	1.60	3.43
Confidence in patient safety management	4.23	0.40	1–5	4.11	4.29
Clinical competence	4.01	0.51	1–5	2.92	4.09
Nursing process	3.47	0.51	1–5	2.30	4.34
Nursing skill	4.47	0.52	1–5	3.34	4.20
Education/cooperation Interpersonal/communication	3.76	0.54	1–5	3.01	4.10
Professional development	4.21	0.53	1–5	3.54	4.50

**Table 3 healthcare-11-02504-t003:** Correlations between alexithymia, confidence in patient safety management, and clinical competence (*n* = 167).

Variables	Alexithymia	Confidence in Patient Safety Management	Clinical Competence
	r (*p*)	r (*p*)	r (*p*)
Alexithymia	1		
Confidence in patient safety management	−0.35 (<0.001)	1	
Clinical competence	−0.35 (<0.001)	0.59 (<0.001)	1

**Table 4 healthcare-11-02504-t004:** Factors influencing clinical competence.

Variables	Model 1			Model 2			Model 3		
	B	β	*p*-Value	B	β	*p*-Value	B	β	*p*-Value
Sex	0.04	0.02	0.790	0.04	0.04	0.607	−0.19	−0.18	0.802
Grade	0.34	0.24	0.007	0.17	0.14	0.065	0.17	0.14	0.063
Scholastic performance	0.13	−0.07	0.331	−0.08	−0.08	0.257	0.02	0.03	0.579
Major satisfaction	−0.20	−0.19	0.046	−0.07	−0.08	0.292	0.03	0.04	0.656
Satisfaction with clinical practice	−0.13	−0.07	0.494	0.08	0.09	0.229	−0.03	−0.04	0.611
Safety education	−0.18	−0.16	0.049	−0.22	−0.13	0.031	−0.20	−0.16	0.022
Religion	0.01	0.01	0.907	0.03	0.030	0.702	−0.01	−0.01	0.974
Subjective health status	−0.21	−0.19	0.027	0.16	−0.13	0.093	−0.07	−0.09	0.236
Alexithymia				−0.53	−0.34	<0.001	0.25	−14	0.049
Confidence in patient safety management							0.54	0.52	<0.001
**R^2^**	0.170			0.154			0.414		
**Adj. R^2^**	0.122			0.143			0.371		
F	3.51			13.14			9.48		
** *p* **	<0.001			<0.001			<0.001		
Durbin–Watson	1.99			2.04			1.97		

Dummy: Sex, Grade, Religion.

## Data Availability

The data sets used and analyzed in the current study are available from the corresponding author on reasonable request. We confirm that this is the case, and ethical considerations or privacy regulations prevent us from sharing the data.

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
