# Peer review of "The Effects of Alexithymia and Confidence in Patient Safety Management on the Clinical Competence of Nursing Students: A Cross-Sectional Survey"

_healthcare, 2023, doi:10.3390/healthcare11182504_

Round 1
Reviewer 1 Report (New Reviewer)
I think this is an interesting study.
Please consider the following
Line 33-34
Competence and skill are not the same thing. Please look up the meaning of the word and use it carefully.
I think it is better to use "competence" instead of "skill."
Line 100-107
This is a redundant expression.
Please state the study aim clearly at the end of the introduction.
It would be better to state the research design only as "cross-sectional descriptive study."
Consider whether abbreviations such as EOF, DDF, or EOT are necessary.
There are many places where there are no spaces between words. Please correct them.
For SD, please indicate that it is the standard deviation and then abbreviate it as SD. Also, I think SD should be expressed as ±SD.
I hope this will be useful to you.
There are many places where there are no spaces between words. It needs to correct them.
Author Response
Reviewer 1:
Line 33-34
Competence and skill are not the same thing. Please look up the meaning of the word and use it carefully.
I think it is better to use "competence" instead of "skill."
Thank you for your good opinion. I modified "competence" instead of "skill."
Line 100-107
This is a redundant expression.
Please state the study aim clearly at the end of the introduction.
At the end of the introduction, I described the purpose of the study as follows.
Therefore, this study aimed to identify the factors influencing the clinical competence of nursing college students. Alexithymia is required for clear communication, and it may have a significant correlation with the confidence of nurses in patient safety management and clinical competence. In the 3rd and 4th year, clinical competence is cultivated through clinical practice. Therefore, in this study, we tried to identify the effect of alexithymia on clinical competence of nursing students in 3rd and 4th grades.
To conduct interventions for helping nursing students in improving their clinical competence, factors that affect clinical competence must be identified. Therefore,the objective of this study was to identify effects of alexithymia and patient safety management on clinical competence of nursing students.
It would be better to state the research design only as "cross-sectional descriptive study."
2.1. Study design described as follows.
The study followed cross-sectional descriptive design.
Consider whether abbreviations such as EOF, DDF, or EOT are necessary.
Thank you for your good opinion. I deleted abbreviations such as EOF, DDF, or EOT
There are many places where there are no spaces between words. Please correct them.
For SD, please indicate that it is the standard deviation and then abbreviate it as SD. Also, I think SD should be expressed as ±SD.
Thank you for your good opinion. I put spaces and corrected them in the entire sentence.
In the case of SD, I marked the standard deviation and then abbreviated it to SD. SD was also modified to ±SD.
Reviewer 2 Report (New Reviewer)
The clinical competence of nursing students is an important area to explore. However, in this study, it is not clear how confidence in patient safety and alexithymia are related to nursing students' clinical competence, as no conceptual framework exists to show the relationship between these variables. Perhaps, the authors may justify the relationship between the study variables based on theoretical support.
In addition, the content in the background is not well organised as the flow of information is not systematic and there is a lot of repetition.
My other concern is regarding the study objectives. The objective in the abstract is “sought to investigate the relationship between alexithymia, confidence in patient safety management and clinical competence among nursing students”, while in the main text aims “to identify the factors influencing the clinical competence of nursing college students” and in the study design to “confirm the factors influencing the clinical competence of nursing students”.
Please ensure that the study objective/aim is consistent though out the manuscript.
Methodology:
Numerous repetition of information was identified.
A total of 90 or 89 items in the questionnaire?
A detailed description of the instruments and their items is not well presented. This is important for readers to understand the content being measured.
Results:
The narrative should avoid repeating what has already been presented in the tables, especially with regard to the general characteristics of the students.
What does the “subjective health status of the students” mean since the result shows that 50 % were in moderate or poor health status?
What do the results in Table 2 show?
Check the sentence accuracy in Lines 271-272 (2 or 4 factors?)
Is there any surprise when fourth graders perform better clinically than third graders? (lines 289-290). Please justify your reasons.
Results should be presented in a logical flow.
Discussion:
The competence reported in this study was based on self-reports. What is the authors’ opinion or comment on this?
Others:
Correct the typos.
Check the referencing style for consistency.
Author Response
The clinical competence of nursing students is an important area to explore. However, in this study, it is not clear how confidence in patient safety and alexithymia are related to nursing students' clinical competence, as no conceptual framework exists to show the relationship between these variables. Perhaps, the authors may justify the relationship between the study variables based on theoretical support.
In addition, the content in the background is not well organised as the flow of information is not systematic and there is a lot of repetition.
Thank you for your good opinion. I described the purpose of the study as follows.
Alexithymia is a disorder in which affected individuals cannot recognize their subjective feelings, effectively control emotions, and have difficulty communicating, particularly when expressing emotions to others and requesting assistance [11]. It is a personality trait where individuals face difficulty in effectively handling and expressing emotions due to a tendency to minimize emotional experience by focusing on the external and not the internal side of themselves, and this symbolizes a defect in the process of recognizing emotions and ability to express emotions [12,13].Nursing students meet several people during clinical practice, where the risk of suppressing emotional expression increases due to increased job tension [14]. In other words, as the inability of nursing students to express their emotions can negatively affect their clinical competence[14],continuous efforts are needed to overcome this issue. Alexithymia in nursing students can lead to difficulties in the form of relationships with patients when performing nursing tasks in the future, as well as a negative impact on their social life [14].
WHO (2019) defines “patient safety” as causing no preventable harm tothe patient during the patient and minimizing the risk of unnecessary harm while caring for their health [15].Confidence in patient safety management is referred to as self-belief in performing tasks related to patient safety. This enables nursing students to participate in voluntary and active clinical practice, thereby improving their clinical competence[16]. Nursing college students, who will play a central role as medical workers, participate in direct and indirect patient care during clinical practice and play an important role in recognizing patient safety problems and improving patient safety after graduation[17].
My other concern is regarding the study objectives. The objective in the abstract is “sought to investigate the relationship between alexithymia, confidence in patient safety management and clinical competence among nursing students”, while in the main text aims “to identify the factors influencing the clinical competence of nursing college students” and in the study design to “confirm the factors influencing the clinical competence of nursing students”.
Please ensure that the study objective/aim is consistent though out the manuscript.
- In abstract and text, the research purpose was modified as follows.
To conduct interventions for helping nursing students in improving their clinical competence, factors that affect clinical competence must be identified. Therefore,the objective of this study was to identify effects of alexithymia and patient safety management on clinical competence of nursing students.
Methodology:
Numerous repetition of information was identified.
After checking the entire paper, I revised the repeated contents.
A total of 90 or 89 items in the questionnaire? A detailed description of the instruments and their items is not well presented. This is important for readers to understand the content being measured.
I modified as follows.
Structured self-report questionnaires comprising86 questions, including 8 on general characteristics, 23 on alexithymia, 10 on confidence in patient safety management, and 45 on clinical competence, was used in this study.
Results:
The narrative should avoid repeating what has already been presented in the tables, especially with regard to the general characteristics of the students.
According to Table 1, this study included 167 nursing students, of whom 123 (73.7%) were female. Regarding grade, 88 participants (52.7%) were in fourth. Regarding academic performance, 99 participants (59.3%) had an average academic score below 3.0. Regarding major satisfaction, 89 participants (53.3%) responded with “satisfaction.” Regarding satisfaction with clinical practice, 86 participants (51.5%) responded with “satisfaction”. Regarding safety education, 146 participants (87.5%) said “Have.”Regarding religion, 118 participants (70.7%) replied “Have”. In terms of subjective health status, 85 participants (50.9%) responded they were “moderate or poor”(Table1).
What does the “subjective health status of the students” mean since the result shows that 50 % were in moderate or poor health status?
The 50.9% of students responded as "moderate or poor." At the follow sentence "What do you think your health is like?"
What do the results in Table 2 show?
The mean score of participants’ alexithymia, confidence patient safety management, clinical competence were Table 2.
Check the sentence accuracy in Lines 271-272 (2 or 4 factors?)
The explanatory power of the two factors was 14.3%.
Is there any surprise when fourth graders perform better clinically than third graders? (lines 289-290). Please justify your reasons.
Our results were similar to those obtained in previous studies [23, 29] that reported that fourth graders exhibit better clinical performance than third graders. We obtained another result that was similar to previous studies [30, 31] that indicated that major satisfaction and clinical practice satisfaction affected clinical competence. The average clinical competence of participant in this study was 4.01 out of 5, which was comparable to that reported in a study (higher than 3.54) [29] that used the same tool as this study to assess third and fourth graders from the same nursing college [30]. Major satisfaction is positively correlated with clinical competence [31]. The improvement in clinical competence observed in this study with an increase in the grade might be attributed to better education and clinical practice. Efforts should be made to increase major satisfaction, as the group with major satisfaction had higher clinical competence than ordinary but dissatisfied groups. In addition, in this study, participants who stated that their subjective health status was good had higher clinical competence than those who said that it wasnormal or bad; therefore, systems and education will be needed to improve the subjective health status of the such individuals.
Results should be presented in a logical flow.
I modified in as logical flow.
Discussion:
The competence reported in this study was based on self-reports. What is the authors’ opinion or comment on this?
I added the following information to the Limitations.
Fourth, in this study, alexithymia, confidence in patient safety management, and clinical competence were measured using self-reported questionnaire only. In the future, it is necessary to collect data through observation or performance evaluation.
Others:
Correct the typos.
I corrected it after checking the typo .
Check the referencing style for consistency.
References have been modified consistently to fit the form.
Reviewer 3 Report (New Reviewer)
An article that deals with an important and relevant topic for healthcare, namely nursing.
I have some suggestions for improvement:
I leave it to your discretion to include in the title some amendment that integrates patient safety. It could be an added value for the visibility of your study.
I suggest that you revise all the writing to avoid errors such as on page 3, line 133, correct: "sentto50% ... ".
If you can select sources as current as possible, please make this change.
Minor editing of English language required
Author Response
An article that deals with an important and relevant topic for healthcare, namely nursing.
I have some suggestions for improvement:
I leave it to your discretion to include in the title some amendment that integrates patient safety. It could be an added value for the visibility of your study.
I revised it as follows.
Title:
Effects of Alexithymia and Confidence in Patient Safety Management on Clinical Competence of Nursing Students: A Cross-Sectional Survey
I suggest that you revise all the writing to avoid errors such as on page 3, line 133, correct: "sentto50% ... ".
I modified it to sent to 50%.
If you can select sources as current as possible, please make this change.
I modified the latest reference.
Reviewer 4 Report (New Reviewer)
Title:
Alexithymia and its relation with clinical Competence is important in your study. Therefore, it is essential to add to your title.
In the introduction, you need to explain how alexithymia and patient safety are related to nursing students' clinical competence.
There is a need to explain more about what it adds to previously published studies.The study needs editing because I found some grammar and spelling mistakes.
Method
It is essential to explain the sampling, sampling size, and response rate in more detail.
Discussion and conclusion
The conclusion section needs to explain in more detail
Author Response
Title:
Alexithymia and its relation with clinical Competence is important in your study. Therefore, it is essential to add to your title.
Thank you for good opinion. I revised it as follows.
Title:
Effects of Alexithymia and Confidence in Patient Safety Management on Clinical Competence of Nursing Students: A Cross-Sectional Survey
Method
It is essential to explain the sampling, sampling size, and response rate in more detail.
Regarding Sampling, I revised and described as follows.
This study used a convenience sampling method to recruit nursing students from two universities in U metropolitan city and G city. The number of samples was determined using the G*Power 3.1 program and confirmed using the following parameters needed for regression analysis: 0.05 significance level, 0.90explanatory power, 0.15medium effect size, and10predictor variables. Following that, we determined that a total of 142 samples were required. Despite a dropout rate of approximately 20%, 170 students were enrolled in this study. Thereafter, 170 copies were collected and 167 people were targeted, excluding 3 who provided insufficient responses. Respose rate was 98.2%.
Discussion and conclusion
The conclusion section needs to explain in more detail
I revised it as follows.
The study participants had relatively high degree of alexithymia, confidence in patient safety management, and clinical competence. In addition, major satisfaction and practical satisfaction were high, and 87.5% had received safety education. The clinical competence according to general characteristics differed according to grade, major satisfaction, practice satisfaction, safety education, and subjective health status perception.
The current study found that students with high levels of confidence in patient safety management and low levels of alexithymia and students who received safety management education had high clinical competence. Confidence in patient safety management had the greatest influence on their clinical competence.
To improve the clinical competence of nursing students, emotional expression inability should be reduced and patient safety management ability should be improved. In addition, safety education before practice is essential to improve clinical competence
This manuscript is a resubmission of an earlier submission. The following is a list of the peer review reports and author responses from that submission.
Round 1
Reviewer 1 Report
The manuscript is generally well- written and clear but needs minor revisions. When reviewing the full paper, I have a few comments:
Line 19-22 stated “The explanatory power of the regression model was 30.5%, which was statistically significant in the study; clinical performance was high in students who were confident in patient safety management and senior nursing students who received education but low in those who suppressed emotional expression.” Not sure the relevancy for this statement in the abstract summary which addresses clinical competence.
Line 81—83 this section addresses the study design and should describe the design only and not the purpose of the study
Line 298—Conclusions. The conclusion should be the last section in the cross-sectional survey and reports on the findings of the paper. The limitations section should precede the conclusion.
Author Response
Reviewer 1:
Line 19-22 stated “The explanatory power of the regression model was 30.5%, which was statistically significant in the study; clinical performance was high in students who were confident in patient safety management and senior nursing students who received education but low in those who suppressed emotional expression.” Not sure the relevancy for this statement in the abstract summary which addresses clinical competence.
The regression model of Model 1 was significant (F = 9.48, p<0.001). Among the general characteristics of the study subjects, grade (β= 0.24, p= 0.007), major satisfaction (β= −0.19, p= 0.046), safety education (β= −0.16, p= 0.049), and subjective health status (β= −0.19, p= 0.027) that had a significant impact on clinical competence. The explanatory power of the four factors was 12.2%. The regression model of Model 2 was significant (F = 3.51, p<0.001). Factors influencing clinical competence were safety education (β= −0.16, p= 0.022), alexithymia (β= −0.14, p= 0.049), and confidence in patient safety management (β= 0.52, p<0.001). The explanatory power of these three factors was 37.1%, and confidence in patient safety management was found to have the greatest influence on the clinical competence. Clinical competence was high among fourth graders, who had high confidence in patient safety management, low alexithymia, and received safety education. Based on these results, it can be inferred that while developing programs to improve clinical competence of nursing students, more attention should be paid to those who scored low in patient safety management and had a high level of alexithymia and safety education should begin in the third grade, i.e., when clinical practice begins.
Line 81—83 this section addresses the study design and should describe the design only and not the purpose of the study
The purpose of this cross-sectional descriptive study was to confirm the factors influencing the clinical competence of nursing students.
Line 298—Conclusions. The conclusion should be the last section in the cross-sectional survey and reports on the findings of the paper. The limitations section should precede the conclusion.
The current study found that students with high levels of confidence in patient safety management and low levels of alexithymia and students who received safety management education had high clinical competence
References
I checked and corrected.
Reviewer 2 Report
Dear Editorial Board,
Thank you for sending the manuscript entitled "Factors Associated with Clinical Competence of Nursing Students: A Cross-Sectional Survey" for review. This manuscript reports the correlation between alexithymia, confidence in patient safety management, and clinical competence among Korean nursing students. In general, I liked the topic of the study, it is interesting since the students’ competency and its determinants or barriers are important in nursing. I think it will be better if the author explains the following details:
1. Abstract:
· Please provide a structured abstract including an introduction or background, aims, methods, results, and conclusion with a reasonable size and important details.
· The result part is started with regression. Please use scientific writing for the result part.
2. Keywords:
· Please provide the most relevant and specialized keywords according to MeSH. For example, Patient safety; Korea; etc.
3. Introduction
· In the introduction, the authors discussed each concept separately. In each paragraph, the authors explain the definition and studies related to the alexithymia or patient safety management concept separately. Since many parameters may be related to nursing students' clinical competence, the authors should state their reasons for choosing these only two parameters. Also, they had better explain the gaps in descriptive studies in these fields.
· In the introduction, the authors have written a few sentences and put several references at the end. This makes it difficult to find which is related to each fact. It is better to refer to each sentence separately and avoid giving references in general. Especially since most of these resources are in Korean, which is not readable for non-Korean speakers. Generally, two-thirds of the 30 articles are in Korean. If it is possible, replace them with Korean studies which have been published in English. I put the example as follows:
· Line 41-44: Please provide a reference for the first sentence. It seems to be one of the references from 7-8.
· Line 55-56: Please provide a reference for the first sentence. It seems to be one of the references from 12-14.
· Line 55-56: What is your mean by “domestic nursing student” vs. “foreign nursing student”? Do you mean Korean vs. other nationalities?
· Did the Korean students have moderate confidence compared with foreigners?
· Line 59-62: Please provide a reference for each fact instead of [10–12, 14–17]. And so on….
· At the end of the introduction, the author’s assumptions are not clearly explained. The readers wouldn’t figure out why they selected these concepts for their survey. In my opinion, the introduction could focus more clearly on the rationale for this study.
4. Method
· Please explain your settings in your country and nursing curriculum:
· How many years do you have in the BSc nursing curriculum? How many semesters?
· And what do you mean by grade? Do you mean the grade is the same as the year?
5. Sampling
· What was the reason that you choose only two universities?
· Why didn't use a more reliable sampling, for example, cluster or categorical sampling?
· Is there only one university in the metropolitan city with a nursing department? If there are several universities, explain the reason for choosing one of them.
· Are the selected universities public? With the same 4-year nursing curriculum?
· Why did you choose the convenient sampling for selecting the students?
· It seems you had the sampling framework that makes it easy to use probability sampling. And how did you control the proportion of men vs. women and grade 3 vs. grade 4?
6. Sample size
· Line 88: The number of participants was calculated according to 7 predictor variables. But you had 14 variables, including 11 general characteristics, alexithymia, confidence in patient safety management, and clinical competence.
7. Measures
· What were the general characteristics? Please name them. There is any information about the general characteristics in Table 1 but sex.
· What was your question about religion, with the yes or no response?
· Line 113: bring past simple verbs instead of the “will be used in this study”.
· Did you consider the following concepts as general characteristics? Major satisfaction, Satisfaction with clinical practice, Subjective health status. And measured each of them with one question for each.
· Line 121: Alexithymia score is on a Likert 5-point scale of 0–4 points. How did you write the range is 1-5 in Table 2?
· What about the questions on confidence and competency? How did you calculate their Likert scales?
8. Data collection
· Considering a total of 90 questions, how did you manage the time to complete the study questionnaires? Was 10-20 minutes enough?
· How did you control the students’ exhaustion? Accompanying in completing the questionnaire? The effect of students' answers on each other?
· Line 154: What is the meaning of “the appropriate subjects in the classroom”
· There is some misunderstanding between the two sections as follows:
Lines 90-101: Despite a dropout rate of about 10%, a total of 180 students were enrolled as study participants. Thereafter, 170 copies were collected and a total of 167 people were targeted, excluding 3 who provided insufficient responses.
And Lines 159-161, a total of 180 questionnaire copies were distributed and collected from those who agreed to participate in this study. However, one with dishonest responses was disqualified. Finally, 167 responses were scrutinized.
9. Data analysis
· Did you test the normality of distribution only for the regression model?
· Why did you use only seven variables in the regression model? If the existence of statistical significance is your reason for selecting the variables, you have to reanalysis the results.
· Which method of data entry is used for Regression?
10. Results
· Please avoid the repetition of the data of the tables in the text.
· The results of the subscales in the Alexithymia, confidence, and competency are not shown.
· Do you test any correlation between Alexithymia and confidence with students’ demographic traits?
11. Tables
· How many students were from a metropolitan university or another university?
· Please avoid the replication of the mean (Sd) scores of the tables in the text.
· Please write the table in an academic format. For example, Table 1: n (%) in a column. Write statistics and p-value instead of t/(p), then explain each statistic sign under the table with a narrow font.
· Table 1: How did you use the t-test for the major satisfaction, satisfaction with clinical practice, and Subjective health status?
· Table 2: Please correct the range of the scores.
· Line 223: what does “alexithymia education” mean?
12. Discussion
· The discussion is not integrated.
· Please use academic format for this part.
· In the first paragraph of the discussion part, write a sum of the study.
· Start each paragraph with one of your findings of the study aims, then write the pros and cons, and finally, a conclusion.
· Most references are in the Korean language, which is not easy to get for foreign readers.
· Please rewrite the strengths of the study.
13. Conclusions
· Line 300: You referred to the low levels of alexithymia. I think the mean score of o2.47 (0.49) of 4 is not low.
· Please rewrite the conclusion according to the reanalyzing of the data.
14. Limitations
· Participation of only third- and fourth-year students is a limitation. As I know, clinical practice is regularly started in the second semester of the first year. So I think if you invited the sophomores, you would find better results.
· Convenience sampling is your study limitation.
15. Reference
· Please recheck references and use EndNote.
· For example, reference 12 needs to modify.
Author Response
- Abstract:
- Please provide a structured abstract including an introduction or background, aims, methods, results, and conclusion with a reasonable size and important details. The result part is started with regression. Please use scientific writing for the result part.
The regression model of Model 1 was significant (F = 9.48, p<0.001). Among the general characteristics of the study subjects, grade (β= 0.24, p= 0.007), major satisfaction (β= −0.19, p= 0.046), safety education (β= −0.16, p= 0.049), and subjective health status (β= −0.19, p= 0.027) that had a significant impact on clinical competence. The explanatory power of the four factors was 12.2%. The regression model of Model 2 was significant (F = 3.51, p<0.001). Factors influencing clinical competence were safety education (β= −0.16, p= 0.022), alexithymia (β= −0.14, p= 0.049), and confidence in patient safety management (β= 0.52, p<0.001). The explanatory power of these three factors was 37.1%, and confidence in patient safety management was found to have the greatest influence on the clinical competence. Clinical competence was high among fourth graders, who had high confidence in patient safety management, low alexithymia, and received safety education. Based on these results, it can be inferred that while developing programs to improve clinical competence of nursing students, more attention should be paid to those who scored low in patient safety management and had a high level of alexithymia and safety education should begin in the third grade, i.e., when clinical practice begins.
- 2. Keywords:
- Please provide the most relevant and specialized keywords according to MeSH. For example, Patient safety; Korea; etc.
: clinical competence; patient safety; students; nursing
- Introduction
- In the introduction, the authors discussed each concept separately. In each paragraph, the authors explain the definition and studies related to the alexithymia or patient safety management concept separately. Since many parameters may be related to nursing students' clinical competence, the authors should state their reasons for choosing these only two parameters. Also, they had better explain the gaps in descriptive studies in these fields.
- In the introduction, the authors have written a few sentences and put several references at the end. This makes it difficult to find which is related to each fact. It is better to refer to each sentence separately and avoid giving references in general. Especially since most of these resources are in Korean, which is not readable for non-Korean speakers. Generally, two-thirds of the 30 articles are in Korean. If it is possible, replace them with Korean studies which have been published in English. I put the example as follows:
Nursing students will have a chance to learn communication skills and methods to communicate with medical staff and subjects in various nursing situations in clinical practice [6]. Nursing students can empathize with patients via efficient communication and relationship formation in clinical practice [7]. However, alexithymia hinders in the development of empathy. Clear communication without expressing emotions is critical in complex and busy clinical field [8], as expressing emotions disrupts clear communication, resulting in patient safety accidents such as crisis discovery, medication errors, delayed diagnosis, and death [9]. Since nursing students must play the role of medical personnel as prospective nurses, it is essential for them to have the expertise and skills to improve the awareness of patient safety and provide safe nursing [10].Therefore, determinants of clinical competence must be identified and reflected in nursing education.
Alexithymia is a disorder in which affected individuals are unable to recognize their subjective feelings, are unable to control emotions effectively, and have difficulty communicating, particularly when expressing emotions to others and requesting assistance [11]. Nursing students have been found to have difficulty expressing their anger and avoid interpersonal relationships when their anger is suppressed [12]. Furthermore, studies have shown that they suppress their anger emotions when alone rather than properly expressing them [13]. Nursing students meet several people during clinical practice, where the risk of suppressing emotional expression increases due to increased job tension [14]. Alexithymia in nursing students can lead to difficulties in the form of relationships with patients when performing nursing tasks in the future, as well as a negative impact on their social life [14].
Confidence in patient safety management is referred to as self-belief in performing tasks related to patient safety. This enables nursing students to participate in voluntary and active clinical practice, thereby improving their performance [15]. Improving confidence in patient safety management enables nursing students to play a competent and professional role as nurses with appropriate clinical skills, knowledge, and attitudes [16].Increased trust in patient safety management can have an impact on clinical competence.
General curriculum of nursing involves gradual sequences. First grade students study general elective subject, and second grade students study basic major subjects and fundamental nursing practices in school. Then, third grade students study advanced major subjects and clinical practices. Clinical practice–based education helps them in acquiring more skills, where they get exposure to various clinical practices, adapt to the role of nursing, and acquire nursing skills [17].
Previous research has found that domestic nursing students about to graduate have lesser knowledge about patient safety than foreign nursing students and have moderate confidence in their abilities[18].Furthermore, one study discovered that although81.6% students received patient safety education, their self-awareness was only moderate [19].Research is being conducted on clinical competence[17], clinical practice satisfaction, clinical stress, patient safety education[19], and communication skills[20, 21], as well as various studies related to subjects affecting patient safety management confidence[18, 22]on nursing students who have had clinical experience. However, current observation-oriented clinical practice has difficulty promoting clinical skills while also limiting clinical performance development through clinical practice [16].
The key to nursing education is to encouragecomplete clinical performance skills among nursing students. Furthermore, modifiable variables that affect nursing students' clinical performance must be identified given the need to ensure clinical competence through nursing education [18].In particular, nursing students may experience clinical maladaptation as a result of alexithymia [13].Furthermore, they acknowledged that they have low confidence in safety management [22], which is an important internal factor that can negatively affect clinical performance. The situation in 2020, when clinical practice in the field is limited, nursing students' confidence in their clinical competence, alexithymia, and safety management confidence may suffer.
- Line 41-44: Please provide a reference for the first sentence. It seems to be one of the references from 7-8.
- Line 55-56: Please provide a reference for the first sentence. It seems to be one of the references from 12-14.
- Line 55-56: What is your mean by “domestic nursing student” vs. “foreign nursing student”? Do you mean Korean vs. other nationalities? · Did the Korean students have moderate confidence compared with foreigners?
Previous research has found that nursing students in Korean about to graduate have lesser knowledge about patient safety than nursing students in other countries
- Line 59-62: Please provide a reference for each fact instead of [10–12, 14–17]. And so on….
Previous research has found that nursing students in Korean about to graduate have lesser knowledge about patient safety than nursing students in other countries and have moderate confidence in their abilities[18].Furthermore, one study discovered that although81.6% students received patient safety education, their self-awareness was only moderate [19].Research is being conducted on clinical competence[17], clinical practice satisfaction, clinical stress, patient safety education[19], and communication skills[20, 21], as well as various studies related to subjects affecting patient safety management confidence[18, 22]on nursing students who have had clinical experience. However, current observation-oriented clinical practice has difficulty promoting clinical skills while also limiting clinical performance development through clinical practice [16].
- At the end of the introduction, the author’s assumptions are not clearly explained. The readers wouldn’t figure out why they selected these concepts for their survey. In my opinion, the introduction could focus more clearly on the rationale for this study.
Therefore, this study aimed to identify the factors influencing the clinical competence of nursing college students in 2020, i.e., during the period when the COVID-19 pandemic situation worsened. Alexithymia is required for clear communication, and it may have a significant correlation with confidence of nurses in patient safety management and clinical performance. Therefore, we would like to confirm the relationship with clinical performance ability through this study.
Therefore, because clinical competence is primarily learned through theory in the first and second grades and cultivated through clinical practice in the third and fourth grades, we attempted to determine the effect of alexithymia and patient safety management on clinical competence.
- Method
- Please explain your settings in your country and nursing curriculum:
General curriculum of nursing involves gradual sequences. First grade students study general elective subject, and second grade students study basic major subjects and fundamental nursing practices in school. Then, third grade students study advanced major subjects and clinical practices. Clinical practice–based education helps them in acquiring more skills, where they get exposure to various clinical practices, adapt to the role of nursing, and acquire nursing skills [17].
- How many years do you have in the BSc nursing curriculum? How many semesters?
There’s 4 grades in BSc nursing curriculum, and 8 semesters for 4 years.
- And what do you mean by grade? Do you mean the grade is the same as the year? ·
Yes.
- Sampling
- What was the reason that you choose only two universities?
We chose them because researchers works in those universities. It’s difficult to get samples of universities when we don’t work at there.
- Why didn't use a more reliable sampling, for example, cluster or categorical sampling?
We had difficulty getting more reliable sampling. Convenience sampling was limitation of our study.
- Is there only one university in the metropolitan city with a nursing department? If there are several universities, explain the reason for choosing one of them.
There are three universities in the metropolitan city with a nursing department. But it’s difficult to get samples of universities when we don’t work at there. So we chose them because researchers works in those universities.
- Are the selected universities public? With the same 4-year nursing curriculum?
No they are private universities but they have 4-year nursing curriculums.
In Korea, all department of nursing have 4-year nursing curriculums.
- Why did you choose the convenient sampling for selecting the students?
In Korea, personal information is managed very strictly, so it’s difficult to get samples of universities if we don’t work at there.
- It seems you had the sampling framework that makes it easy to use probability sampling. And how did you control the proportion of men vs. women and grade 3 vs. grade 4?
Thank you for your great opinion. Probability sampling is much better, but we used convenience sampling because of difficulty of permission of survey. In 2020, rate of male nursing students is about 20% in Korea, so we tried to control the proportion the rate over 20%. And we tried to make the proportion of grade 3 and grade 4 as same rates.
- Sample size
- Line 88: The number of participants was calculated according to 7 predictor variables. But you had 14 variables, including 11 general characteristics, alexithymia, confidence in patient safety management, and clinical competence.
8 general characteristics(sex, grade, Scholastic performance, Major satisfaction, Satisfaction with clinical practice, Safety education, Religion, Subjective health status
This study used a convenience sampling method to recruit nursing students from two universities in U metropolitan city and G city. The number of samples was determined using the G*Power 3.1 program and confirmed using the following parameters needed for regression analysis: 0.05significance level, 0.90 explanatory power, 0.15 medium effect size, and 10predictor variables
- Measures
- What were the general characteristics? Please name them. There is any information about the general characteristics in Table 1 but sex.
- What was your question about religion, with the yes or no response?
Do you have a religion?
- Line 113: bring past simple verbs instead of the “will be used in this study”.
“will be used in this study”.à would be used in this study.
- Did you consider the following concepts as general characteristics? Major satisfaction, Satisfaction with clinical practice, Subjective health status. And measured each of them with one question for each.
- Line 121: Alexithymia score is on a Likert 5-point scale of 0–4 points. How did you write the range is 1-5 in Table 2?
We revised it to 0-4.
- What about the questions on confidence and competency? How did you calculate their Likert scales
There were questions on confidence that ask the students for their confidence of 10 factors about patient safety management.
For each answers for each questions, we gave Likert scales,
Very confident – 5
Confident – 4
Moderate – 3
Not confident – 2
Not confident at all - 1
There were 45 questions on competency that ask the students for their ability of nursing process, nursing skill, cooperation, communication, and professional development.
For each answers for each questions, we gave Likert scales,
Strongly agree – 5
Agree – 4
Neutral – 3
Disagree – 2
Strongly disagree - 1
- Data collection
- Considering a total of 90 questions, how did you manage the time to complete the study questionnaires? Was 10-20 minutes enough?
I think it is enough to respond the survey.
- How did you control the students’ exhaustion? Accompanying in completing the questionnaire? The effect of students' answers on each other?
We gave some rewards (like some pens) for students. The students did survey individually so they didn’t effected each other.
- Line 154: What is the meaning of “the appropriate subjects in the classroom”
- There is some misunderstanding between the two sections as follows:
Lines 90-101: Despite a dropout rate of about 10%, a total of 180 students were enrolled as study participants. Thereafter, 170 copies were collected and a total of 167 people were targeted, excluding 3 who provided insufficient responses.
. Following that, we determined that a total of 153 samples were required. Despite a dropout rate of about 10%, a total of 180 students were enrolled as study participants. Thereafter, 180 copies were collected and a total of 167 people were targeted, excluding 13 who provided insufficient responses.
And Lines 159-161, a total of 180 questionnaire copies were distributed and collected from those who agreed to participate in this study. However, one with dishonest responses was disqualified. Finally, 167 responses were scrutinized.
- Data analysis
- Did you test the normality of distribution only for the regression model?
No, we tested the normality of distribution for regression and t-test.
- Why did you use only seven variables in the regression model? If the existence of statistical significance is your reason for selecting the variables, you have to reanalysis the results.
We analysised again with 10 variables in the regression model.
- Which method of data entry is used for Regression?
Entry
- Results
- Please avoid the repetition of the data of the tables in the text.
- The results of the subscales in the Alexithymia, confidence, and competency are not shown.
- Do you test any correlation between Alexithymia and confidence with students’ demographic traits?
I checked and revised.
- Tables
- How many students were from a metropolitan university or another university?
- Please avoid the replication of the mean (Sd) scores of the tables in the text.
- Please write the table in an academic format. For example, Table 1: n (%) in a column. Write statistics and p-value instead of t/(p), then explain each statistic sign under the table with a narrow font.
I checked and revised.
- Table 1: How did you use the t-test for the major satisfaction, satisfaction with clinical practice, and Subjective health status?
- Table 2: Please correct the range of the scores
- Line 223: what does “alexithymia education” mean?
alexithymia
- Discussion
- The discussion is not integrated.
- Please use academic format for this part.
- In the first paragraph of the discussion part, write a sum of the study.
- Start each paragraph with one of your findings of the study aims, then write the pros and cons, and finally, a conclusion.
- Most references are in the Korean language, which is not easy to get for foreign readers.
- Please rewrite the strengths of the study.
I checked and revised.
This study was conducted to identify factors that affect the clinical competence of nursing students. The first regression model identified general characteristics affecting the clinical competence of nursing students. Fourth graders had higher clinical competence than third graders, and subjects who reported having major satisfaction were more satisfied than those who reported having moderate or dissatisfaction. Subjects who were satisfied with clinical practice scored higher than those who were moderately or dissatisfied with clinical practice. Subjects with good health status outperformed those with moderate or poor health status in terms of clinical performance.
Our results were similar to those obtained in previous studies [23, 29] that reported that fourth graders exhibit better clinical performance than third graders. Also, there was another similar result to previous studies [30, 31] that reported that major satisfaction and clinical practice satisfaction affected clinical competency.
The average clinical competence of participant in this study was 4.01 out of 5, which was comparable to that reported in a study(higher than 3.54)[29] that used the same tool as this study to assess third and fourth graders from the same nursing college [30]. Major satisfaction is positively correlated with clinical competence [31]; therefore, it is necessary to improve satisfaction of students in the major and clinical practice as well as enhance health status to improve the clinical competence of nursing students.
In the second regression model, alexithymia and confidence in patient safety management were added to the general characteristics that affect the clinical competency of nursing students. The current study evaluated whether nursing students’ clinical competence was affected by confidence in patient safety management, safety education, alexithymia.
The identification of a relationship between alexithymia and clinical competence in the absence of research on alexithymia of nursing college students is of great importance. This study is significant for the following reasons. First, because this study was designed for third- and fourth-grade students, the degree of variability in nursing students can be predicted. Thus, by developing and implementing clinical competence programs, nursing students' clinical competence can be improved. Such programs should be offered to nursing students with low confidence in patient safety management or a high alexithymia score. Second, because we discovered that safety management education influences clinical competence, it is critical to strengthen safety management education for nursing students before clinical practice.
- Conclusions
- Line 300: You referred to the low levels of alexithymia. I think the mean score of o2.47 (0.49) of 4 is not low.
- Please rewrite the conclusion according to the reanalyzing of the data.
The current study found that students with high levels of confidence in patient safety management and low levels of alexithymia and students who received safety management education had high clinical competence.
- Limitations
- Participation of only third- and fourth-year students is a limitation. As I know, clinical practice is regularly started in the second semester of the first year. So I think if you invited the sophomores, you would find better results.
Thank you for your opinion. In korea, First grade students study general elective subject, and second grade students study basic major subjects and fundamental nursing practices in school. Then, third grade students study advanced major subjects and clinical practices. So most research of clinical competence targets on third and fourth-year students. I will try to include second year students in next research.
- Convenience sampling is your study limitation.
Yes, convenience sampling is our study limitation
- Reference
- Please recheck references and use EndNote.
- For example, reference 12 needs to modify.
I checked and corrected.